# Association of center-level operative volume and acute outcomes following robotic-assisted colectomy for colorectal cancer

Nam Yong Cho[1], Shineui Kim[1], Joseph Hadaya[1], Nikhil Chervu[1], Shayan Ebrahimian[1], Emma Cruz[1], Hanjoo Lee[2], Peyman Benharash[1,3]*

1 Cardiovascular Outcomes Research Laboratories (CORELAB), Division of Cardiac Surgery, David Geffen School of Medicine at UCLA, Los Angeles, California, United States of America, 2 Division of Colorectal Surgery, UCLA-Harbor Medical Center, Los Angeles, California, United States of America, 3 Department of Surgery, Division of Cardiac Surgery, David Geffen School of Medicine at UCLA, Los Angeles, California, United States of America

* pbenharash@mednet.ucla.edu

## Abstract

### Background

The adoption of robotic-assisted colectomy (RAC) remains limited due to high costs. There is a paucity of data regarding the impact of institutional robotic experience on costs in patients undergoing RAC for colorectal cancer.

### Methods

All adult patients undergoing RAC for colorectal cancer were identified using the 2016–2020 Nationwide Readmissions Database. A multivariable regression to model major adverse events (MAE) was developed with the inclusion of institutional robotic surgery volume as restricted cubic splines. The volume corresponding to the inflection point of the spline was used to stratify hospitals into high- (HVH) or low-volume (LVH). We subsequently examined the association of HVH status with costs, length of stay (LOS), MAE, non-home discharge and 30-day unplanned readmission.

### Results

Among the 39,064 patients undergoing RAC, 65.2% were treated at HVH. Following risk adjustment, RAC at HVH was associated with reduced index hospitalization costs by $2,000 (95%CI $1,500−2,400) and LOS by 0.3 days (95%CI 0.2–0.5 days) as well as decreased odds of MAE (AOR 0.86, 95%CI 0.77–0.96). Both non-home discharge and 30-day unplanned readmission were not associated with hospital volume. In our cross-volume analysis, we found an increase in institutional RAC and overall robotic volumes to be associated with reduced odds of MAE.

**Data availability statement:** This study uses the Nationwide Readmissions Database (NRD) which is a part of Nationwide Healthcare Cost & Utilization Project (HCUP) maintained by the at the Agency for Healthcare Research and Quality (AHRQ). The authors of this study did not have special access privilege to the data that others would not have. However, because the NRD database is maintained by AHRQ, data can be accessed directly from their website at (https://www.hcup-us.ahrq.gov/) upon completion of Data Use Agreement for researchers who meet the criteria for access to confidential data. Data cannot be provided directly by the authors of this study due to required specific approval from the AHRQ.

**Funding:** The author(s) received no specific funding for this work.

**Competing interests:** The authors have declared that no competing interests exist.

## Conclusion

The present study demonstrated higher institutional robotic-assisted operation volume to be associated with reduced MAE and cost in patients undergoing RAC. The findings suggest the potential benefits of increasing expertise and implementing efficient practices in robotic-assisted surgery programs.

## Introduction

Robotic-assisted surgery has been increasingly adopted in the United States and is now utilized in >15% of all general surgical procedures [1]. Compared to laparoscopy, robotic-assisted surgery allows for improved visualization, tremor control and greater range of motion given the use of wristed instruments. This technology has been employed in robotic-assisted colectomy (RAC), with its safety and feasibility widely established in the literature [2–5]. Prior work has highlighted the cost-effectiveness of the robotic-assisted colectomy despite similar average costs compared to the laparoscopic approach [6] (costs: laparoscopic $26,114, robotic $26,859, open $31,714). Nonetheless, utilization of RAC in the United States is reported to be low, in part, due to the significant costs associated with acquisition and maintenance of robotic equipment [7–9]. Despite maturation of robotics and the potential for enhanced recovery, Simianu et al. reported RAC to remain less cost-effective compared to laparoscopy in the current era [6].

Prior work has revealed higher center-specific robotic surgical volume to be associated with lower costs for various cancer operations including lobectomy and pancreaticoduodenectomy [10,11]. Specifically, an increase in the overall robotic-assisted caseloads have been shown to reduce operative times and postoperative length of stay [6]. However, there is a paucity of data regarding the definition of high-volume for RAC as well as its associated cost-volume outcome relationship. Moreover, whether this observation is mediated through RAC specific or overall robotic volume remains unclear. In the present study, we examined the association of institutional robotic-assisted surgery volume with clinical outcomes and various measures of resource use in a national cohort of patients receiving RAC. We hypothesized greater institutional robotic-assisted operation volume to be associated with reduced mortality, complications, length of stay and 30-day unplanned readmissions.

## Materials and methods

This was a retrospective study of the 2016−2020 Nationwide Readmissions Database (NRD). The NRD is an all-payer readmissions database that provides accurate estimates for ~ 59% of all hospitalization across the United States using survey methodology. The NRD tracks readmissions across hospitals within each calendar year. Using relevant *International Classification of Disease, 10th Revision* (ICD-10) codes, all adult (≥18 years) hospitalizations entailing elective RAC (right hemicolectomy, transverse colectomy, left hemicolectomy, sigmoid colectomy and total colectomy)

for benign or malignant colorectal neoplasms were identified (S1 Table). Records missing data for age, sex and charges, were excluded from analysis (0.01%; Fig 1).

Patient and hospital characteristics, including age, sex and zip code-based income level, were defined using the HCUP data dictionary [12]. The van Walraven modification of the Elixhauser Comorbidity Index [13] was used to numerically capture the burden of chronic conditions. Other clinically relevant comorbidities and complications were tabulated using ICD-10 codes [14]. Hospitalization costs were calculated by application of cost-to-charge ratios to total hospitalization charges and subsequent adjustment for inflation using the 2020 Personal Health Index [15].

The overall hospital robotic case volume was calculated as the number of overall robotic-assisted operations performed at each institution each year. Similarly, institutional robotic-assisted colectomy volume was determined as the number of robotic-assisted colectomy cases performed yearly. As the NRD does not track hospitals across calendar years, caseload estimates were calculated independently for each year in the study. A logistic regression model for major adverse events (MAE) was developed with overall robotic-assisted operation case volume modeled as a restricted cubic spline. Of note, MAE was defined as a composite of in-hospital mortality and neurologic (stroke or transient ischemic attack), cardiovascular (cardiac arrest, myocardial infarction, ventricular arrhythmias, or tamponade), respiratory (acute respiratory failure, acute respiratory distress syndrome, prolonged ventilation greater than 96 hours, pneumonia, or pneumothorax), thromboembolic (deep venous thrombosis or pulmonary embolism), infectious (sepsis or surgical site infection) and gastrointestinal (anastomotic leak, gastrointestinal bleeding, gastrointestinal obstruction) complications.

We utilized institutional robotic-assisted colectomy volume as a continuous variable in the spline analysis to evaluate the relationship between procedural volume and MAE. To identify the inflection point, we calculated the global maximum of the second derivative in the spline between volume and MAE. Institutions with volumes at or above this inflection point (N = 207) were classified as high-volume hospitals (HVHs), while those below the threshold were designated as low-volume hospitals (LVHs). The primary outcome of the study was index hospitalization costs, while secondary outcomes included MAE, postoperative length of stay (LOS), non-home discharge and 30-day unplanned readmission.

Categorical variables are expressed as group proportions, while continuous variables are shown as medians with interquartile range (IQR). Comparisons by volume status were made using Pearson's $\chi^2$-test for categorical and the Mann-Whitney U test for continuous variables. Entropy balancing was utilized to generate a weighted comparison group with similar distribution of covariates. Superior to propensity matching, this methodology [16] seeks a set of sample weights that satisfy balance constraints while maintaining the entire cohort for analysis. Multivariable linear and logistic

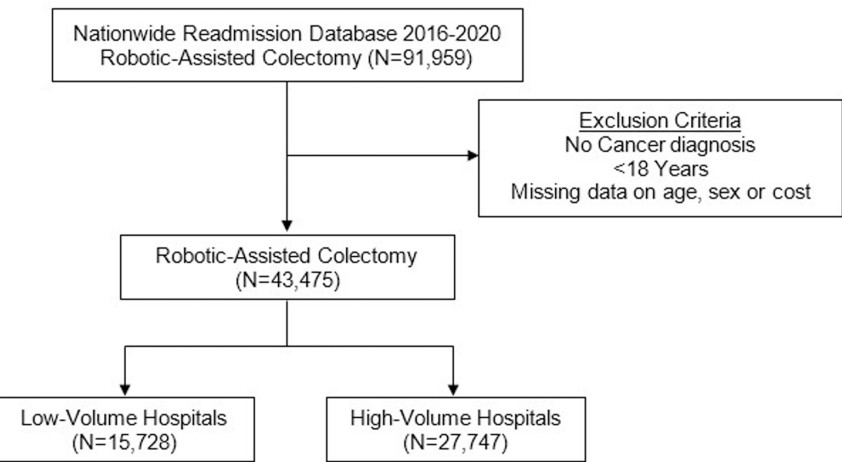

**Fig 1. Flow diagram of study cohort and survey-weighted sample size.**

regressions were subsequently developed to evaluate the association of HVH status with the outcomes of interest. Regression outputs are shown as adjusted odds ratios (AOR) or beta coefficients (β), both with 95% confidence intervals (CI). To account for effect size, standardized mean difference (SMD) >0.10 was utilized to demonstrate statistical significance. Data was gathered and reported according to the Strengthening the Reporting of Observational studies in Epidemiology (STROBE) guideline. All statistical analyses were performed using Stata 16.0 (StataCorp, College Station, TX). Data visualization was achieved using Python (Python Software Foundation) package *Matplotlib*. Due to the fully de-identified nature of the NRD, this study was deemed exempt from full review by the Institutional Review Board at the University of California, Los Angeles, Los Angeles, CA (IRB:17-001112, approved July 26, 2017).

## Results

### Study cohort baseline characteristics at HVH and LVH

Of an estimated 39,064 patients undergoing elective robotic-assisted colectomy, 65.2% received care at HVH. Compared to LVH, patients at HVH were younger (64 [54–73] vs 66 [56–74] years, SMD = 0.12) and often from highest income quartile (29.2 vs 24.4%, SMD = 0.15). Patients at both LVH and HVH had a similar distribution of female sex and comparable Elixhauser Comorbidity score. In addition, patients at HVH more frequently underwent RAC at urban teaching (87.6 vs 70.7%, SMD = 0.18) and large-bed hospitals (66.2 vs 51.4%, SMD = 0.34) compared to those at LVH. As shown on Table 1, patients undergoing RAC at LVH had similar comorbidities including diabetes (22.1 vs 20.5%, SMD = 0.04), congestive heart failure (3.6 vs 3.0%, SMD = 0.04) and malnutrition (2.9 vs 2.9%, SMD = 0.02).

### Outcomes at HVH and LVH

On unadjusted analysis, patients undergoing RAC at HVH had lower index hospitalization costs ($21,600 [16,500–29,100] vs $22,800 [16,900–31,800]) compared to those at LVH. In addition, compared to those at LVH, patients at HVH had lower rates of MAE (6.0 vs 7.0%, SMD = 0.12). Patients undergoing RAC at HVH had lower hospital duration of stay (3 [2–5] vs 4 [3–5] days, SMD = 0.13) while rates of 30-day unplanned readmission (7.6 vs 7.9%, SMD = 0.01) were similar compared to LVH (Table 2). Following RAC, patients at HVH had lower rates of non-home discharge (4.4 vs 5.3%, SMD = 0.02) than others.

Following risk adjustment, RAC at HVH was associated with reduced index hospitalization costs by $2,000 (95%CI $1,500−2,400). Factors associated with an increment in hospitalization costs included insurance status with Medicaid (β $3,400, 95%CI $2,400−4,300), highest income quartile (β $5,000, 95%CI $4,300−5,600) and large hospital bed size (β $2,800, 95%CI $2,100−3,500). Conversion to laparotomy was also associated with increased hospitalization costs by $17,700 (95%CI $14,000−21,400). On our spline analysis, HVH demonstrated lower likelihood of MAE compared to LVH, as shown in Fig 2. Management at HVH was not associated with reduced odds of mortality (AOR 0.82, 95%CI 0.49–1.37) or odds of conversion to open procedures (AOR 1.02, 95%CI 0.64–1.64) compared to LVH. Furthermore, HVH status was associated with a 0.2-day decrement in LOS (95%CI 0.1–0.4 days). Both non-home discharge and 30-day unplanned readmission and were not associated with hospital volume (Table 3).

### Cross-volume analysis

Following adjustment for patient and operative characteristics, the relationship between institutional RAC and overall robotic operation volumes was analyzed in regard to MAE. As shown in Fig 3, both an increase in institutional RAC and overall robotic operation volumes were found to be associated with reduced odds of MAE.

## Discussion

In the present study, we examined the impact of institutional robotic-assisted operation volume on various clinical and financial endpoints following RAC for colorectal cancer. We found that increasing overall robotic-assisted operation

**Table 1. Demographic of patients undergoing robotic colectomy stratified by management at stratified by low- (LVH) and high-volume hospital (HVH) designation. SMD, Standardized Mean Difference.**

|  | LVH (N = 13,607) | HVH (N = 25,457) | SMD |
|---|---|---|---|
| Age (year) | 66 [56–74] | 64 [54–73] | 0.12 |
| Female Sex (%) | 46.7 | 47.0 | 0.01 |
| Elixhauser Index Score | 3 [2–4] | 3 [2–4] | 0.05 |
| Hospital Bed Size (%) |  |  | 0.34 |
| Small | 18.4 | 9.0 |  |
| Medium | 30.2 | 24.8 |  |
| Large | 51.4 | 66.2 |  |
| Hospital Teaching Status (%) |  |  | 0.18 |
| Rural | 6.4 | 1.3 |  |
| Urban Nonteaching | 22.9 | 11.2 |  |
| Urban Teaching | 70.7 | 87.6 |  |
| Insurance Status (%) |  |  | 0.09 |
| Medicare | 50.8 | 46.7 |  |
| Medicaid | 7.0 | 5.1 |  |
| Private | 39.0 | 45.4 |  |
| Self-pay | 1.1 | 1.1 |  |
| Income quartile (%) |  |  | 0.15 |
| 0–25th | 23.5 | 18.7 |  |
| 25th–50th | 26.0 | 25.2 |  |
| 50th–75th | 26.1 | 26.9 |  |
| 75th–100th | 24.4 | 29.2 |  |
| Anemia (%) | 6.5 | 6.9 | 0.01 |
| Diabetes (%) | 22.1 | 20.5 | 0.04 |
| Heart Failure (%) | 3.6 | 3.0 | 0.04 |
| Malnutrition (%) | 2.9 | 2.9 | 0.01 |
| Obesity (%) | 19.8 | 19.6 | 0.01 |
| Smoking (%) | 10.3 | 9.7 | 0.02 |

**Table 2. Unadjusted analysis of clinical and financial outcomes in patients undergoing robotic-assisted colectomy stratified by low- (LVH) and high-volume hospital (HVH) designation. D, Days. Comparison of categorical variables is reported using 95% confidence intervals while that of continuous variables are shown with median and interquartile range.**

|  | LVH (N = 15,728) | HVH (N = 27,747) | SMD |
|---|---|---|---|
| Major Adverse Event (%) | 7.0 | 6.0 | 0.12 |
| Mortality (%) | 0.3 | 0.3 | 0.02 |
| Conversion to Open (%) | 0.4 | 0.4 | 0.01 |
| Cost ($1,000) | 22.8 [16.9–31.8] | 21.6 [16.5–29.1] | 0.15 |
| LOS (d) | 4 [3–5] | 3 [2–5] | 0.13 |
| Readmission in 30 Days (%) | 7.9 | 7.6 | 0.01 |
| Nonhome Discharge (%) | 5.3 | 4.4 | 0.02 |

volume to be associated with a non-linear decrease in major adverse events. Additionally, high overall robotic-assisted operative volume was associated with decreased index hospitalization costs following RAC in our analysis. Notably, patients who underwent RAC at HVH experienced lower rates of MAE and shorter hospital duration of stay. We performed

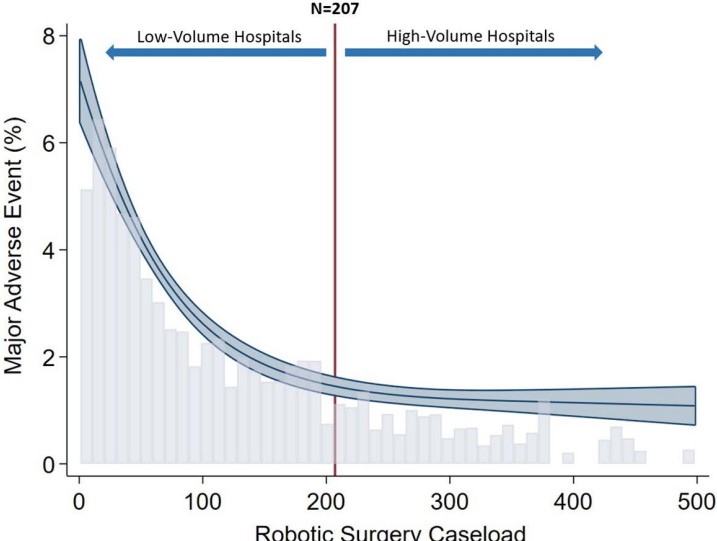

**Fig 2. Spline analysis of risk-adjusted major adverse event (MAE) following robotic-assisted colectomy stratified by overall robotic operation caseload.**

**Table 3. Risk-adjusted analysis of clinical and financial outcomes in patients undergoing robotic-assisted colectomy at high-volume hospitals. D, Days. Estimates are reported as adjusted odds ratio (AOR) for dichotomous outcomes and β-coefficients for continuous outcomes with corresponding 95% confidence interval for both (Reference: low-volume hospital).**

|  | Estimates (β/AOR) | 95%CI |
|---|---|---|
| Major Adverse Event | 0.90* | 0.80–0.97 |
| Mortality | 0.82 | 0.49–1.37 |
| Conversion to Open | 1.02 | 0.64–1.64 |
| Cost ($1,000) | −2.0* | −2.4–−1.5 |
| LOS (d) | −0.2* | −0.4–−0.1 |
| Readmission in 30 Days | 0.97 | 0.91–1.04 |
| Nonhome Discharge | 0.96 | 0.82–1.12 |

*; P<0.05

a cross-volume analysis and found overall robotic volume and RAC volume to have a synergistic effect. Specifically, as the volume of operations increased, there was a corresponding decrease in the incidence of MAE. These findings warrant further discussion.

Previous studies have reported high overall robotic-assisted surgical volume to be associated with reduced costs for cancer surgery, including lobectomy, pancreaticoduodenectomy and prostatectomy [17,18]. In the present work on RAC, we found that an increase in the number of all robotic operations performed at a center was associated with decreased index hospitalization costs. This finding may be explained by the fact that centers with a high robotic operation volume also tend to be high-volume colectomy institutions. While adoption community centers with low annual colectomy caseloads may have financial disadvantages, high-volume hospitals can likely afford robotic platforms. Furthermore, institutions with higher-volume colectomy volume may have enhanced integrated networks [19] providing comprehensive cancer services or are academic centers with high rates of referral for RAC. At such centers, prior studies have consistently

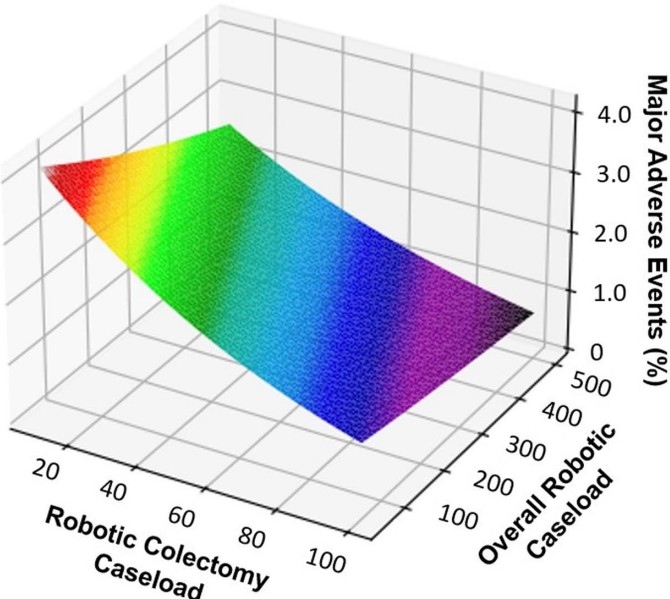

**Fig 3. Surface plot demonstrating the relationship of institutional robotic colectomy caseload and overall robotic operation caseload with major adverse events.**

demonstrated an inverse relationship between institutional surgical volume and operative time across various procedures including laparoscopic colectomy, which may ultimately reduce cost [20–22]. Other significant factors contributing to cost reduction include the implementation of streamlined practices [23] at both hospital and operating room levels. Notably, improving operating room turnover efficiency and fostering comprehensive operating room staff training to enhance proficiency with the robotic platform are critical measures to lower costs. Current cost reduction strategies primarily focus on minimizing instrument redundancies and enhancing operative efficiency, both of which are facilitated by increasing surgeon proficiency through training and experience [24]. In order to develop a more cost-effective framework for robotic-assisted operations, further investigation on driving factors of cost in RAC is warranted.

One potential mechanism for reducing inpatient costs is the reduced rates of MAE at high-volume RAC centers. Consistent with our study, Keller et al. analyzed 18 months of a national inpatient database, demonstrating both high RAC case and surgeon volumes to be associated with decreased perioperative complications [25]. Additionally, high robotic-assisted operative volumes were associated with lower rates of postoperative ileus and anastomotic leak, as well as shorter hospital stays and lower costs following RAC. These findings are substantiated by several factors, including improved infrastructure and shorter learning curves observed in high-volume hospitals specifically for RAC [26–28]. We hypothesize that center-specific experiences of surgeons at high-volume RAC hospitals are the main drivers of decreasing complication rates [29]. A prior study evaluating the outcomes of colorectal surgery for rectal cancer has found the lowest complication rate [30] among patients operated on by high-volume surgeons. Undoubtedly, shortened learning curve by the surgeon and the operating room staff for RAC contributes to decreased rates of perioperative complications and, consequently, reduced costs [31,32].

Despite the rapid expansion of robotic surgery, we noted the majority of the RAC to be performed at low volume institutions. This finding emphasizes the need for streamlined robotic surgery infrastructures even in low volume hospitals. For instance, Swedish hospitals were able to maintain equal, sometimes better outcomes than Dutch hospitals despite having lower volume and later adoption of robotic surgery. The authors attributed this to the standardized training

programs and higher per-surgeon volume in Sweden [29], which led to a shorter learning curve. Appropriate patient selection during the early phases of the adoption of a robotic program has been proposed to lower MAE and cost. In an Austrian study evaluating the implementation of a robotic surgery program, MAE was particularly pronounced in complex cases during the early phases of enrollment. The authors recommended proceeding with simple cases and gradually building up to cases with higher complexity [30] as the robotic program matures. In order to improve infrastructure and shorten the learning curve at LVHs, these approaches can effectively support the objectives of value-based healthcare delivery.

The adoption of robotic surgery programs has been met with considerable debate, with challenges stemming from the inherent difficulties of converting robotic surgery to laparotomies and the high costs [27] associated with maintaining the device. Nonetheless, a recent Canadian study from a large tertiary center showed that the hospitalization cost trended downward [28] after the robotic colorectal program was implemented. The cost of robotic surgery was effectively mitigated through rigorous training of operating room staff and utilization of established postoperative care pathways [32], which led to a decrease in laparotomy conversion rates. Our findings complement this by suggesting the strategic distribution of robotic technology across various operations may be a crucial structural factor that makes robotic surgery programs successful. Appropriate allocation of robotic operations to centers with high-case volumes and skilled surgeons will ensure familiarity with procedures and treatment pathways [33] throughout the hospital system. For example, in the Netherlands, the number of hospitals performing robotic rectal cancer surgeries decreased between 2012 and 2018 [29] despite an overall increase in the volume of overall surgical operations. This was achieved by centralizing care at high-volume centers. However, such a national-level paradigm shift may not be feasible in the U.S., where healthcare delivery is still largely privatized and centralization may impair access to care [34]. Nonetheless, adoption of minimally invasive further studies are necessary to establish safe and cost-effective protocols for robotic surgery as it continues to advance in technology.

The present study has several noteworthy limitations. Due to the administrative nature of the NRD, our access to detailed clinical data, including laboratory values and oncologic staging for patients undergoing RAC, is limited. Additionally, the NRD lacks the capability to evaluate institutional surgeon and center experience on robotic-assisted colectomy as well as their learning curve over the study period. Moreover, the absence of hospital region information in the NRD hinders our ability to assess geographic variations in costs. Notably, the charge data provided by the NRD does not offer a detailed breakdown of operating room costs or long-term expenditures associated with the robotic system or acquisition at particular institutions. We also could not assess the institutional capacity to monitor and provide prehabilitation or enhanced recovery pathways after surgery. As we defined high RAC volume based on MAE, our specific cut-off may not be applicable to other hospital-level performance. Despite these limitations, we used appropriate methodologies to show the relationship between robotic-assisted operative volume and inpatient costs at a large scale.

## Conclusions

In this nationwide retrospective study, we examined the association of overall hospital robotic-assisted operative volume on clinical and financial outcomes following RAC for colorectal cancer. We found increased robotic-assisted operation volume to be significantly associated with decreased index hospitalization costs, lower rates of MAE and shorter LOS. Additionally, our study showed a synergistic effect between overall robotic-assisted operation volume and the institutional caseload of robotic-assisted colectomies in reducing the odds of MAE. These findings highlight the potential benefits that may derive from increased institutional expertise and the implementation of streamlined practices of robotic-assisted surgery. In light of the evolving landscape of value-based healthcare delivery, our study may supplement healthcare leaders in making informed decisions regarding the incorporation of robotic-assisted surgery into their practices.

## Supporting information

**S1 Table. International Classification of Disease, Ninth and Tenth revision, Codes (ICD-9/10) for robotic operations and colectomy procedures.**
(DOCX)

## Author contributions

**Conceptualization:** Nam Yong Cho, Nikhil Chervu, Hanjoo Lee, Peyman Benharash.

**Formal analysis:** Nam Yong Cho, Shineui Kim.

**Methodology:** Nam Yong Cho, Joseph Hadaya, Hanjoo Lee, Peyman Benharash.

**Software:** Shayan Ebrahimian.

**Supervision:** Hanjoo Lee, Peyman Benharash.

**Validation:** Joseph Hadaya.

**Visualization:** Emma Cruz.

**Writing – original draft:** Nam Yong Cho.

**Writing – review & editing:** Shineui Kim, Joseph Hadaya, Nikhil Chervu, Shayan Ebrahimian, Hanjoo Lee, Peyman Benharash.

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
