## [Decision Letter · Decision Letter 0]

PONE-D-24-04448Association of Center-Level Operative Volume and Acute Outcomes following Robotic-Assisted Colectomy for MalignancyPLOS ONE

Dear Dr. Benharash,

Thank you for submitting your manuscript to PLOS ONE. After careful consideration, we feel that it has merit but does not fully meet PLOS ONE’s publication criteria as it currently stands. Therefore, we invite you to submit a revised version of the manuscript that addresses the points raised during the review process.

We look forward to receiving your revised manuscript.

Kind regards,

Sakarie Mustafe Hidig, MD

Academic Editor

PLOS ONE

Journal Requirements:

Reviewers' comments:

Reviewer's Responses to Questions

**Comments to the Author**

1. Is the manuscript technically sound, and do the data support the conclusions?

Reviewer #1: No

Reviewer #2: Yes

Reviewer #3: Yes

2. Has the statistical analysis been performed appropriately and rigorously? 

Reviewer #1: No

Reviewer #2: Yes

Reviewer #3: Yes

3. Have the authors made all data underlying the findings in their manuscript fully available?

Reviewer #1: No

Reviewer #2: Yes

Reviewer #3: Yes

4. Is the manuscript presented in an intelligible fashion and written in standard English?

Reviewer #1: Yes

Reviewer #2: Yes

Reviewer #3: Yes

5. Review Comments to the Author

Reviewer #1: This is a retrospective review using the Nationwide Readmission Database looking at costs and major adverse events with the use of the robot for colectomy based on volume.

I was wondering if this database covers years current (such as 2020-2023)? If so, why were they not included.

It was really unclear how costs were calculated in this study-making the conclusions difficult to interpret.

I am not sure that the Nationwide Readmissions database is the most reliable database to use for this analysis. Especially since only a little over 50% of hospitalizations have accurate information.

In the methodology it indicates that both benign and malignant neoplasms data was pulled from, but the entire manuscript indicates malignancy.

Why were only colectomies included and not proctectomies? That is where the real benefit of the robot of colorectal surgery has been seen.

I don’t think calculating high versus low based on overall robotic operations at the institution each year is appropriate.

Given that most of the procedures at LVH were not elective I don’t think looking at costs or MAE is accurate given these are usually sicker patients.

Don’t understand the statement of prehab and rates of liver mets- not accurate and not supported by that reference.

Reviewer #2: 1. When you use an abbreviation in the text for the first time, it is recommended to write the full name and follow it immediately with an abbreviation. RAC in the abstract section.

2. How were high and low-volume hospitals defined in your study?

3. Please revise your reference on the costs of colectomy surgery in the introduction section (the cost of colectomy open surgery nearly doubled that of laparoscopic and robotic surgery?)

4. You mentioned in your study that the operating room level may contribute to low cost, I suggest you write this point more clearly for your readers to understand.

5. You seem to indicate that prehabilitation protocols contribute to cost reduction, can you elaborate more?

Reviewer #3: According to the authors, the Nationwide Readmissions Database (NRD), a component of the Nationwide Healthcare Cost & Utilization Project (HCUP) run by the Agency for Healthcare Research and Quality (AHRQ), is where the data used to support the study's conclusions may be found. However, researchers who meet the requirements for access to secret data must complete a Data Use Agreement in order to access this data. The authors cannot immediately disclose the data because they require explicit consent from AHRQ, and they do not have any special access credentials that others would not have. As a result, even while the data is available, there are limitations that prevent complete access.

Another statistical test that could be used in addition to cross-volume analysis is multivariable logistic regression. This test is particularly useful for examining the relationship between one or more independent variables (such as institutional robotic surgery volume, patient demographics, and comorbidities) and a binary dependent variable (such as the occurrence of major adverse events or readmissions).

Thus the use of ICT could enhance the data collection and thus it could make the research in ease and more variables could be identified.

6. PLOS authors have the option to publish the peer review history of their article (what does this mean? ). If published, this will include your full peer review and any attached files.

**Do you want your identity to be public for this peer review?** For information about this choice, including consent withdrawal, please see our Privacy Policy .

Reviewer #1: No

Reviewer #2: No

Reviewer #3: No

---

## [Author Response · Author response to Decision Letter 1]

15 Sep 2024

Reviewer #1: This is a retrospective review using the Nationwide Readmission Database looking at costs and major adverse events with the use of the robot for colectomy based on volume.

I was wondering if this database covers years current (such as 2020-2023)? If so, why were they not included.

Thank you for this comment. Recent years were unavailable due to the data collection and processing time required by Agency for Healthcare Research and Quality.

It was really unclear how costs were calculated in this study-making the conclusions difficult to interpret. I am not sure that the Nationwide Readmissions database is the most reliable database to use for this analysis. Especially since only a little over 50% of hospitalizations have accurate information.

Thank you for your comment. The Nationwide Readmission Database (NRD) is the largest database that tracks inpatient hospitalization costs, using discharge data from 30 geographically dispersed states, representing 59.6% of all U.S. hospitalizations. While some bias may occur due to coding errors, the NRD is widely regarded as a reliable and accurate data source for the hospitalizations it includes. By comparison, the National Inpatient Sample (NIS) provides cost information but only represents a 20-percent stratified sample of all discharges from U.S. hospitals. This makes the NRD’s 59.6% coverage a more accurate and comprehensive sample.

In the methodology it indicates that both benign and malignant neoplasms data was pulled from, but the entire manuscript indicates malignancy.

We appreciate this comment. Manuscript wordings have been modified accordingly.

Why were only colectomies included and not proctectomies? That is where the real benefit of the robot of colorectal surgery has been seen.

Thank you for your comment. In our database, we identified ~1000 patients undergoing robotic proctectomy for benign or malignant neoplasms of the colon, sigmoid junction, or rectum each year. Due to the relatively low volume of robotic prostatectomies, it was likely that these procedures were concentrated in highly specialized centers with higher surgical volumes. As a result, they were not a primary focus of our study.

I don’t think calculating high versus low based on overall robotic operations at the institution each year is appropriate. Given that most of the procedures at LVH were not elective I don’t think looking at costs or MAE is accurate given these are usually sicker patients.

Thank you for this important point. To ensure a better comparison group, we narrowed our analysis to elective operations only. Following the exclusion of non-elective patients, management at LVH was associated with greater odds of MAE and incremental cost.

Don’t understand the statement of prehab and rates of liver mets- not accurate and not supported by that reference.

We appreciate your point. Given that studies looking into prehabilitation with a reduction in hospitalization costs for cancer patients undergoing robotic surgeries are sparse in the literature and not supported by the data presented, we removed the sentence. We have added the following line to our limitation:

“We also could not assess the institutional capacity to monitor and provide prehabilitation or enhanced recovery pathways after surgery.” (Page 15)

Reviewer #2:

1. When you use an abbreviation in the text for the first time, it is recommended to write the full name and follow it immediately with an abbreviation. RAC in the abstract section.

Thank you for this comment. We have revised our abstract accordingly.

2. How were high and low-volume hospitals defined in your study?

We appreciate your comment. We have added the following lines to the method section to further clarify our definition of high- and low- volume centers.

“We utilized institutional robotic-assisted colectomy volume as a continuous variable in the spline analysis to evaluate the relationship between procedural volume and MAE.

To identify the inflection point, we calculated the global maximum of the second derivative in the spline between volume and MAE. Institutions with volumes at or above this inflection point (N=207) were classified as high-volume hospitals (HVHs), while those below the threshold were designated as low-volume hospitals (LVHs).” (Page 6)

3. Please revise your reference on the costs of colectomy surgery in the introduction section (the cost of colectomy open surgery nearly doubled that of laparoscopic and robotic surgery?)

Thank you very much for this point. We have added the following reference to the manuscript [1]. Similar to their study, we found open operations were approximately 120% of the cost of laparoscopic and robotic approaches.

1. Simianu VV, Gaertner WB, Kuntz K, et al. Cost-effectiveness Evaluation of Laparoscopic Versus Robotic Minimally Invasive Colectomy. Ann Surg. 2020;272(2):334. doi:10.1097/SLA.0000000000003196

4. You mentioned in your study that the operating room level may contribute to low cost, I suggest you write this point more clearly for your readers to understand.

We appreciate your comment. We have edited our discussion section to further elaborate on operating room level interventions that may contribute to lower cost.

“Notably, improving operating room turnover efficiency and fostering comprehensive operating room staff training to enhance proficiency with the robotic platform are critical measures to lower costs. Current cost reduction strategies primarily focus on minimizing instrument redundancies and enhancing operative efficiency, both of which are facilitated by increasing surgeon proficiency through training and experience.24,25 In order to develop a more cost-effective framework for robotic-assisted operations, further investigation on driving factors of cost in RAC is warranted.” (Page 13).

5. You seem to indicate that prehabilitation protocols contribute to cost reduction, can you elaborate more?

Thank you for this comment. Previous studies have shown that the prehabilitation pathway, with or without enhanced recovery after surgery protocol, may reduce the risk of bleeding complications, hospital duration of stay, and cost. [1] In patients with cancer, prehabilitation improves functional outcomes in the patients after cancer resection operations [2]. Nonetheless, given our database is limited to inpatient data, we removed the sentence regarding cost reduction as our data cannot support the claim.

1. Ploussard G, Almeras C, Beauval J, et al. A combination of enhanced recovery after surgery and prehabilitation pathways improves perioperative outcomes and costs for robotic radical prostatectomy. Cancer. 2020;126(18):4148-4155. doi:10.1002/cncr.33061

2. Coderre D, Brahmbhatt P, Hunter TL, Baima J. Cancer prehabilitation in practice: the current evidence. Curr Oncol Rep. 2022;24(11):1569-1577. doi:10.1007/s11912-022-01304-1

Reviewer #3:

According to the authors, the Nationwide Readmissions Database (NRD), a component of the Nationwide Healthcare Cost & Utilization Project (HCUP) run by the Agency for Healthcare Research and Quality (AHRQ), is where the data used to support the study’s conclusions may be found. However, researchers who meet the requirements for access to secret data must complete a Data Use Agreement in order to access this data. The authors cannot immediately disclose the data because they require explicit consent from AHRQ, and they do not have any special access credentials that others would not have. As a result, even while the data is available, there are limitations that prevent complete access.

We appreciate this comment. We will make sure to acknowledge this point when we submit the manuscript.

Another statistical test that could be used in addition to cross-volume analysis is multivariable logistic regression. This test is particularly useful for examining the relationship between one or more independent variables (such as institutional robotic surgery volume, patient demographics, and comorbidities) and a binary dependent variable (such as the occurrence of major adverse events or readmissions).

Thus the use of ICT could enhance the data collection and thus it could make the research in ease and more variables could be identified.

Thank you very much for this comment. We have used multivariable logistic and linear regression to evaluate the association of high-volume hospital status with outcomes of interest as suggested.

---

## [Decision Letter · Decision Letter 1]

Association of Center-Level Operative Volume and Acute Outcomes following Robotic-Assisted Colectomy for Colorectal Cancer

PONE-D-24-04448R1

Dear Dr. Peyman Benharash

We’re pleased to inform you that your manuscript has been judged scientifically suitable for publication and will be formally accepted for publication once it meets all outstanding technical requirements.

Kind regards,

Sakarie Mustafe Hidig, MD

Academic Editor

PLOS ONE

Reviewers' comments:

Reviewer's Responses to Questions

**Comments to the Author**

1. If the authors have adequately addressed your comments raised in a previous round of review and you feel that this manuscript is now acceptable for publication, you may indicate that here to bypass the “Comments to the Author” section, enter your conflict of interest statement in the “Confidential to Editor” section, and submit your "Accept" recommendation.

Reviewer #1: (No Response)

Reviewer #2: All comments have been addressed

Reviewer #3: All comments have been addressed

2. Is the manuscript technically sound, and do the data support the conclusions?

Reviewer #1: Yes

Reviewer #2: Partly

Reviewer #3: Yes

3. Has the statistical analysis been performed appropriately and rigorously? 

Reviewer #1: Yes

Reviewer #2: Yes

Reviewer #3: (No Response)

4. Have the authors made all data underlying the findings in their manuscript fully available?

Reviewer #1: No

Reviewer #2: Yes

Reviewer #3: Yes

5. Is the manuscript presented in an intelligible fashion and written in standard English?

Reviewer #1: Yes

Reviewer #2: Yes

Reviewer #3: Yes

6. Review Comments to the Author

Reviewer #1: I think the authors have addressed all of the reviewers' comments except for reviewer 3's first comment. Is there consent required? Has it been obtained.

Reviewer #2: the manuscript has improved significantly; therefore, it can be published as it is. Congratulations!

Reviewer #3: By analyzing the connection between patient outcomes in robotically assisted colectomy and center-level operative volume, the study tackles a crucial topic in surgical oncology. This is especially critical as healthcare systems are progressively implementing robotic technology, and knowledge of how surgical volume affects these systems can help direct training and budget allocation which is the highlight of this manuscript and it is properly explained.

This research could ultimately lead to improved patient care and reduced complications.

The use of multivariable logistic regression, enhances the credibility of the findings. This methodological rigor is essential for establishing causal relationships and should be highlighted as a strength of the study.

The research also underscores the importance of surgical training and experience in robotic-assisted procedures.

All the other comments were addressed but a little improvement in the fashion of writing could improve the paper more better.

7. PLOS authors have the option to publish the peer review history of their article (what does this mean? ). If published, this will include your full peer review and any attached files.

**Do you want your identity to be public for this peer review?** For information about this choice, including consent withdrawal, please see our Privacy Policy .

Reviewer #1: No

Reviewer #2: No

Reviewer #3: No

---

## [Editor Report · Acceptance letter]

PONE-D-24-04448R1

PLOS ONE

Dear Dr. Benharash,

I'm pleased to inform you that your manuscript has been deemed suitable for publication in PLOS ONE. Congratulations! Your manuscript is now being handed over to our production team.

Kind regards,

on behalf of

Dr. Sakarie Mustafe Hidig

Academic Editor

PLOS ONE